# Food Risk Analysis: Towards a Better Understanding of “Hazard” and “Risk” in EU Food Legislation

**DOI:** 10.3390/foods12152857

**Published:** 2023-07-27

**Authors:** Ana-Andreea Cioca, Livija Tušar, Tomaž Langerholc

**Affiliations:** 1Department of Microbiology, Biochemistry, Molecular Biology and Biotechnology, Faculty of Agriculture and Life Sciences, University of Maribor, 2311 Hoče, Slovenia; tomaz.langerholc@um.si; 2Department of Biochemistry and Molecular and Structural Biology, Jožef Stefan Institute, Jamova Cesta 39, 1000 Ljubljana, Slovenia; livija.tusar@ijs.si; 3Centre of Excellence for Integrated Approaches in Chemistry and Biology of Proteins (CIPKeBiP), Jamova Cesta 39, 1000 Ljubljana, Slovenia

**Keywords:** risk analysis, risk communication, hazard, risk, food regulation, food safety

## Abstract

For risk communication, it is important to understand the difference between “hazard” and “risk”. Definitions can be found in Codex Alimentarius and the European Union (EU) General Food Regulation (EC) No. 178/2002. The use of these terms as synonyms or their interchange is a recurrent issue in the area of food safety, despite awareness-raising messages sent by EFSA (European Food Safety Authority) and other interested entities. A quick screening of the EU’s food regulations revealed several inconsistencies. Hence, it was considered necessary to further investigate if regulations could act as a source for this problem. A software tool was developed to support the detection and listing of inconsistent translations of “hazard” and “risk” in certain EU food regulations. Subsequently, native-speaking experts working in food safety from each EU country were asked to provide their individual scientific opinion on the prepared list. All data were statistically analysed after applying numerical scores (1–5) describing different levels of consistency. Results showed that the most common problem was the interchange of “hazard” with “risk” and vice versa. This lack of consistency can create confusion that can further translate into misjudgments at food risk assessment and communication levels.

## 1. Introduction

Food safety remains a public health priority and a global responsibility of the governments of EU countries in their politics and long-term strategies [1,2,3]. Moreover, food safety is intensively discussed in the current context of climate change [4,5,6,7,8]. In an effort to minimise the contribution to global greenhouse gas emissions (GHG), food systems are undergoing some radical changes that will directly affect the field of food safety [9,10,11,12,13]. The move toward more sustainable food and agriculture presents new challenges not only for food risk assessors but also for managers and the general public, which is composed of consumers [14]. Before the new challenges of future food systems can be addressed by redesigning food safety assessment strategies, it is essential to review and improve certain aspects that still affect the understanding of ‘food risk analysis’ as a whole [15]. Of particular importance in this case is risk communication and, concomitantly, a clear understanding of the specific terminology used in risk assessment and management [16,17].

The terms “hazard” and ”risk” in the context of food safety first appeared more than 20 years ago in the Codex Alimentarius, then in Regulation (EC) No. 178/2002 [18] and other European Union regulations that are part of food law. Unfortunately, the distinction between these two has always been problematic [19]. There is a minimal number of studies on this topic, of which very few are related to food safety. In an empirical study, Scheer et al. (2014) [20] concluded that “hazard” and “risk” are perceived very differently depending on the perspective of the stakeholders, regardless of their definition in the Codex Alimentarius [21]. Moreover, an online survey by Wiedemann et al. (2010) [22] revealed that numerous people cannot distinguish between the two notions but rather mix hazard and risk aspects. In studies which are not related to food safety, the same issue appears, indicating the exercise of differentiating between “hazard” and “risk” is not an easy one. In 2020, Freudenstein et al. [23], concluded that risk communication needs to develop means for empowering the public to differentiate between “hazards” and “risks” after an online survey on the topic of radiofrequency electromagnetic fields (RF EMF) and health. In the subsequent years, the European Food Safety Authority (EFSA) has repeatedly urged a distinction between the two terms [24,25].

In accordance with the Codex Alimentarius, the General Food Law Regulation (EC) No. 178/2002, adopted by the European Parliament and the Council, defined the two terms as follows:➢“Hazard” is a biological, chemical or physical substance in or a condition of a food or feed that may have an adverse effect on health;➢“Risk” is a function of the probability of an adverse health effect and the severity of that effect resulting from a hazard.

These definitions are also reproduced in Regulation (EU) 2017/625 [26] with minor changes that do not affect the intended meaning, as follows:➢“Hazard” means an agent or condition that may have potentially harmful effects on human, animal or plant health, animal welfare or the environment;➢“Risk” is a function of the likelihood of an adverse health effect on human, animal, or plant health, animal welfare, or the environment and the severity of that effect resulting from a hazard.

In the European Community (EC), the definitions of “hazard” and ”risk” should be uniformly adopted in national legislation. The message conveyed by legal documents must be unambiguous and leave no room for interpretation [27]. Risk communication in this form should minimise consumer confusion, promote understanding of risk assessment and risk management, and increase public confidence in food quality and safety measures [28]. Currently, food regulations adopted in English by the EC are translated into national languages; hence translations can be inconsistent. This is because either there are no appropriate expressions in other languages to distinguish between ”hazard” and ”risk” [29,30] or simply because translators themselves misunderstand the terminology.

The aim of this study is to identify, compare, and critically analyse the terms “hazard” and “risk” in the original English versions of some important food regulations (EU) 2002/178; (EU) 2004/852 [31]; (EU) 2004/853 [32]; (EU) 2017/625; (EU) 2019/1381 [33] and their equivalents in the official EU languages. We also compare the most recent regulation with the oldest in terms of the number of discrepancies confirmed by us and by experts. The analysis of the current state of official regulations will allow policy-makers to plan further awareness-raising activities in EU Member States in order to introduce correct terminology. Furthermore, the methodology described could be adapted and applied horizontally, i.e., also to regulations from other areas, such as animal health (i.e., Regulation (EU) 2016/429) [34].

## 2. Materials and Methods

### 2.1. Selection of Regulations

Several EU food regulations were screened to find the most relevant to food safety, which include risk assessment terminology. These are, for example, “hazard”, “risk”, “hazard analysis”, “risk assessment”, and “risk analysis”. Based on this criterion and their undeniable importance in practice, the following food regulations were selected: Regulation (EU) 2002/178 on general principles and requirements of food law, Regulation (EU) 2004/852 on food hygiene, Regulation (EU) 2004/853 on specific hygiene rules for food of animal origin, Regulation (EU) 2017/625 on official controls and other official activities performed to ensure the application of food and feed law, rules on animal health and welfare, plant health and plant protection products, and the single version of Regulation (EU) 2019/1381 on the transparency and sustainability of EU risk assessment in the food chain. All regulations are publicly available on EUR-Lex, the official website for European Union law [35]. With the exception of Regulation (EU) 1381/2019, which was available in a single version, an original act and the latest versions were extracted for each regulation. The most recent versions or new versions at the time of the search were as follows: 26/05/2021 version of Regulation (EU) 2002/178, 24/03/2021 version of Regulation (EU) 2004/852, 28/10/2021 version of Regulation (EU) 2004/853, 28/10/2021 version of Regulation (EU) 2017/625). In the comparison between the old and new versions of the document, the so-called legal act was considered the old version. The purpose of this comparison was to examine improvements in the translation of “hazard”, “risk”, and related constructions, such as “hazard analysis”,” risk assessment”, and” risk analysis”.

### 2.2. Software Development

A software tool HA-RI was developed in Java (Spring Framework) to compare two regulations, one in English and one in another language, to find inconsistent translations of “hazard” and “risk” or other structures related to food risk assessment: risk assessment, risk analysis, hazard analysis. For more details on the development and features of HA-RI, see Appendix A. The tool was the solution for the quick screening of multiple food regulations and their document versions on the basis of advance-determined inconsistencies presented in Appendix A. The tool was developed for individual use, but permission to access it can be granted to users upon request. 

#### Creation of Inconsistency Tables for the EU Languages

The input data for the software tool HA-RI were the first (in this case English) and second language, the URL address of the selected legislation for the first and second language and the words for “hazard” and “risk” for both selected languages (Appendix A). The form for entering the data into the program is shown in Figure 1. Figure 2 shows the part of the output document in PDF format where all inconsistencies and consistencies are marked in red and yellow, respectively. Finally, the tables of inconsistencies were created in Excel in a form suitable for statistical analysis and graphical representation. These results were called “our determination of inconsistencies”.

### 2.3. Input from Food Experts

In January 2022, lists of inconsistencies in legislation due to translations were formulated. The first invitation to experts who were supposed to comment on these inconsistencies was sent to native speakers of various official EU languages. The reason for requesting their assistance was the fact that we, the authors of the presented study, are native speakers of Romanian (RO) and Slovenian (SL) and might, thus, have different opinions when analysing inconsistencies in the translation of “hazard” and “risk” between English and other official EU languages, as semantic freedom prevails in native languages. The validation or invalidation of our results was therefore considered useful and crucial for the interpretation of the final result. The opinions of the experts were important for us to confirm the interchanges and have contributed a great deal to the harmonisation of the analysis carried out.

The request was addressed to experts with different backgrounds (e.g., veterinarians, food technologists, nutritionists, biologists, etc.) trained or active in food risk assessment and communication. The expert group was composed of EU-FORA (Food Risk Assessment Programme, EFSA) Cycle 2021–2022, EU-FORA alumni from previous cohorts, representatives from EFSA Focal Points, and other proposed experts. One expert with native language in the following main EU languages accepted our invitation: Bulgarian (BG), Czech (CS), Danish (DA), German (DE), Estonian (ET), Greek (EL), French (FR), Croatian (HR), Latvian (LV), Lithuanian (LT), Hungarian (HU), Dutch (NL), Polish (PL), Slovak (SK), Slovenian (SL), Finnish (FI), and Swedish (SV). For some languages, like Spanish, Portuguese and Romanian, 2 native experts accepted, while for Italian, 3 native experts accepted. No expert in Maltese was available at the time of the study. We performed the inconsistencies assessment for all 23 EU languages, and our opinion was the only one given for Romanian (RO) and Slovene (SL) inconsistencies. In total, the opinions of 27 experts for 22 EU languages were included in the present study.

### 2.4. Analysis

First, the experts provided synonyms for “hazard”, “risk”, and related structures in their own languages (Appendix A). Second, the selected regulations were reviewed by HA-RI to ensure independent identification of inconsistencies. The act in English of each selected regulation was compared with the corresponding version in BG, CS, ES, DA, DE, ET, EL, FR, HR, IT, LV, LT, HU, NL, PL, PT, RO, SK, SL, FI, SV. The same was applied to the document versions mentioned in Section 2.1.

After screening, the results were recorded, and the surveys were prepared as tables in Excel for the native experts. The inconsistencies to be discussed with the native experts were very specific and related to particular paragraphs of food legislation where deviations from the correct translation were found. For each inconsistency, the exact location in the legislation was provided by page and paragraph so that it could be easily verified in the case. An observation in the form of an objective question to the native speaker (e.g., ‘Is “hazard” translated as “risk”?’, etc.) to obtain his or her response was provided (Appendix A). Different combinations of terms lead to selected inconsistencies, e.g., H-R is when “risk” is used instead of “hazard” (Table 1).

Our opinion was not included in the survey to obtain an objective response from the experts. 

To quantify the results, a scoring system for the responses was introduced. The quantitative codes ranged from 1 (consistent translation) to 5 (inconsistent translation), as explained in Table 2. The inconsistencies found by us were marked with 5. Our definition of terms was based on Regulation (EU) 2002/178 and Regulation (EU) 2017/625 (with almost identical definitions), as well as synonyms previously provided by the experts (e.g., for the term “hazard” in Spanish (ES) our reference was “peligro” used in Regulation (EU) 2017/625, also agreed by native experts, instead of “factor de peligro” used in Regulation (EU) 2002/178; for the term “hazard” in Slovak (SK) our reference was “nebezpečenstvo” used in Regulation (EU) 2017/625, also agreed by native experts, instead of “ohrozenie” used in Regulation (EU) 2002/178)) (Appendix A). The experts had the freedom to rate these inconsistencies on a scale of 1 to 5 (Appendix A). Inconsistencies with a score of 1, 2 and 3 were still treated as consistent translations, while those with a score of 4 or 5 were treated as expert-confirmed inconsistencies. It is important to note that a score of 3 (expert unsure) was extremely rare. If more than one expert responded to the same language, the individual responses were not compared, but their common reply was the average value of their individual scores. 

First, the results were to show the total number of inconsistencies between the old and the new version of the regulations. Only the inconsistencies identified by us and confirmed by the experts were considered. To obtain a clear idea of where these agreed inconsistencies are located at the level of the regulations, a pivot table was created (Appendix A). 

Second, the analysis of the results allowed the comparison between all subfamilies/groups of EU languages (Table 3) in terms of overall inconsistencies. Finally, correlations were made using the criteria in Table 4.

The vast majority of EU languages belong to the Indo-European language family. The three main subfamilies are Germanic, Romance, and Slavic. The others are Baltic, Celtic, Hellenic and Uralic. Maltese is a special language. It is a Central Semitic language, derived from late medieval Sicilian Arabic with Romance overlays, spoken by the Maltese people. It is the national language of Malta and the only official Semitic language in the European Union.

### 2.5. Statistical Analysis

Given the ordinal variables, Spearman’s rank correlation coefficient was calculated. For this purpose, the variables were coded (Table 1, Table 2, Table 3 and Table 4). The variable ‘’agreement’’ was introduced to determine the agreement between our assessment and the experts (Table 2 and Table 3). Statistical calculations about expert-confirmed inconsistencies were performed for the old version of Regulations (EU) 2002/178, 2004/852, 2004/853, 2017/625, the new version of the same regulations and the regulation (EU) 2019/1381. The relationship was considered significant when the *p* value was less than 0.05.

Comparisons between selected ordinal variables were performed using the Kruskal–Wallis test. The statistically significant difference was confirmed when the *p* value was less than 0.05.

### 2.6. Clustering

Finally, the clustering of languages by the number of inconsistencies was performed. The selected variables were: R-H, H-RF, RA-RD, H-SD, and H-R (shown in Table 1). After optimisation, we selected the hierarchical Ward method for clustering, except for Regulation (EU) 1381/2019, where the average and the centroid hierarchical method (Figure A3) were used.

The Cubic Clustering Criterion (CCC) [36] was used to determine the number of clusters. When CCC was greater than 2, the number of clusters was relevant (groups of languages that are well separated). The clusters were represented by dendrograms.

Codes written in SAS for Windows were used for the graphical representations (bars and pies), comparison tests, clustering, and dendrograms [37]. 

## 3. Results

To perform the analysis of ‘’hazard’’ and ‘’risk’’ use in existing food legislation, a six-step procedure was followed (Figure 3). Upon selection of relevant regulations, languages and experts, the HA-RI software tool was applied to create an initial list of inconsistencies for selected regulations and languages. These lists of inconsistencies were evaluated by experts and us.

The English language was the language of reference. The Gaelic language had no inconsistencies. The total number of the selected five types of inconsistencies (R-H, H-RF, RA-RD, H-SD, H-R, explained in Table 1) that we identified was 657, of which 361 (54.95%) were in the older version, 288 (43.84%) in the newer version, and 8 (1.21%) in Regulation (EU) 2019/1381. A total of 610 inconsistencies, or 92.84%, were agreed between us and experts, of which 336 (55.08%) were in the older version, 267 (43.77%) in the newer version, and 7 (1.15%) in Regulation (EU) 2019/1381 (Table 5).

In most cases, our observations on the inconsistencies and awarded scores (Table 2) were confirmed by the native experts. However, there were a few cases where the experts found the use of “hazard” and “risk” as synonyms acceptable for reasons of linguistic freedom (Table 3 and Table 5). For example, for the Slovak language for the new regulation, four inconsistencies were identified, but only two were confirmed by the experts (Table 5). 

The substitution of the word “hazard” with the word “risk” was most frequently observed in the Italic group, followed by the Slavic, Germanic and Baltic groups (Figure 4). In other language groups, such as Hamito-Semitic, Uralic and Hellenic, the mentioned substitution was rare.

Substitution of the word “risk” with the word “hazard” was more common in Slavic, Uralic, Baltic, Italian and Germanic languages (Figure 5). Maltese from the Hamito-Semitic group had a small number of substitutions, Greek from the Hellenic group none.

Other substitutions discovered in the terms “risk” and “hazard” were also H-SD, H-RF and RA-RD (Figure 6). Greek showed that “hazard” was often substituted with “risk factor” and “source of danger”. Lithuanians from the Baltic group also showed a high number of inconsistencies based on the substitution of “hazard” with “risk factor”, while in Slovenians from the Slavic group, six interchanges were detected. In Spanish, a language from the Italic group, “risk assessment” was substituted with “risk determination”.

### 3.1. Number of Inconsistencies in the Old Version of Regulations No. 2002/178, 2004/852, 2004/853 and 2017/625

The highest number of inconsistencies identified by consensus between our and the experts’ assessment in the old version of regulations was found for LT (48), EL (39), and for SL (31); for the other, the number of inconsistencies was less than 30 (Figure 7a). The lowest number was found and confirmed for SV (1) and SK (1).

Figure A1a in Appendix B shows the number of inconsistencies in old versions of regulations No. 2002/178, 2004/852, 2004/853 and 2017/625 found by our (author) assessment. 

The inconsistency rate (the number of interchanges in relation to the total number of occurrences of certain terms) varied both between languages and between the terms analysed (Appendix A). The R-H interchange rate peaked for Lithuanian (13.8%) and Polish (11.1%). The rate of “hazard” interchanges (H-R + H-RF + H-SD) peaked in Greek (55.7%), followed by Romanian (32.9%), Slovenian (31.4%) and Danish (30%). The interchange RA-RD was only present in Spanish and was interchanged in 34.8% of the cases.

### 3.2. Number of Inconsistencies in the New Version of Regulations No. 2002/178, 2004/852, 2004/853 and 2017/625

The highest number of inconsistencies identified by consensus between our and the experts’ assessment of the new version of regulations was for EL (36), followed by LT (25) and SL (25); for the others, the number of inconsistencies was less than 25 (Figure 7b). The lowest number was found and confirmed for SV (1).

The number of expert-confirmed inconsistencies in the new version of Regulations (EU) 2002/178, 2004/852, 2004/853, and 2017/625 was 20.5% lower than in the old version of the same regulations, which is also reflected in inconsistency rates (Appendix A). The R-H interchange rate peaked for Polish (6.1%), Croatian (5.1%) and Lithuanian (4.4%). The rate of “hazard” interchanges (H-R + H-RF + H-SD) even increased in Greek (57.1%) compared to the old versions (55.7%), followed by Portuguese (31.7%), Romanian (28.6%) and Slovenian (28.6%). The interchange RA-RD was again only present in Spanish and was interchanged in 54.2% of the cases (an increase from 34.8% in the old versions).

Figure A1b in Appendix B shows the number of inconsistencies in new versions of regulations No. 2002/178, 2004/852, 2004/853 and 2017/625 found in our assessment.

### 3.3. Number of Inconsistencies in the Single Regulation (EU) No. 2019/1381

Regulation No. 2019/1381 was the newest by the date of publication. Not surprisingly, the number of inconsistencies identified by consensus between our and the experts’ assessment per language was very low (the highest was 2) (Figure 8) compared to the regulations with old and new versions. In addition, we found one inconsistency in Hungarian. However, the national experts did not agree on this case. Figure A2 in Appendix B shows the number of inconsistencies identified by the authors. 

Since there was a low number of interchanges in the single Regulation (EU) No. 2019/1381, inconsistency rates decreased. The R-H interchange rate in German and Finnish was 2.6% and 1.3% in Estonian. The rate of “hazard” interchanges (H-R + H-RF + H-SD) in Greek was 100%, while the interchange RA-RD was not present (Appendix A).

### 3.4. Correlations between the Variables

In the old/new versions of regulations No. 2002/178, 2004/852, 2004/853 and 2017/625, as well as in the single regulation No. 2019/1381, some relationships among variables, for example, ‘’agreed scores between authors and native food experts’’, ‘’language groups’’, ‘’different languages’’ and ‘’accession years’’, were statistically significant based on the results of Spearman rank correlation coefficients calculations. The Spearman rank correlation coefficient calculations are presented in Table A1, Table A2 and Table A3 in Appendix B.

In the case of old versions of Regulations No. 2002/178, 2004/852, 2004/853 and 2017/625, the relationships between “different languages” or “language groups” with “evaluators” in general (*p* = 0.0381) and “the year of the state’s accession to the EU” with “groups of languages” or “language groups” (*p* < 0.0001) were the main factors determining the number of inconsistencies found (in this case R-H, H-RF, RA-RD, H-SD, and H-R). In the new versions of these regulations, the same relationships were still open except for “groups of languages” with “evaluators”.

In the case of the single Regulation EU No. 2019/1381, no statistically significant correlations were found for the above-listed variables.

### 3.5. Comparisons by Using the Kruskal–Wallis Test

Comparison of the type of authors/experts who agreed on inconsistencies between old/new Regulations No. 2002/178, 2004/852, 2004/853 and 2017/625, and Regulation 1381 revealed a statistically significant difference (*p* = 0.03). 

### 3.6. Clusters

In the old versions of Regulations EU No. 2002/178, 2004/852, 2004/853 and 2017/625, language EL was in a separate cluster (green) due to the highest number of inconsistencies (Figure 9). The group of languages ET, HU, HR, PL, and LT belonged to the cluster (brown) with a percentage of R-H substitutions above 65% and a total number of inconsistencies between 10 and 48. Languages CS, PT, RO, DA, and SL formed the cluster (red) with a proportion of H-R over 61% and a total number of inconsistencies between 19 and 31. The cluster of language groups consisting of FR, NL, and IT (inside the blue) had a proportion of H-R over 90%, but the number of inconsistencies was low—between 10 and 13. The remaining languages of the old versions were also placed in the blue cluster, which had a smaller number of R-H and H-R type inconsistencies than the second and third clusters (from 1 to 9). 

In the new versions of Regulations EU No. 2002/178, 2004/852, 2004/853 and 2017/625, language EL was again in the separate cluster (green) due to the highest number of inconsistencies. The brown cluster of inconsistencies included HR, HU, PL, and LT, with R-H exceeding 52% and ranging from 16 to 25 inconsistencies. The cluster (red) consisted of CS, IT, DA, RO, PT and SL, with an H-R percentage of over 60% and a total number of inconsistencies ranging from 14 to 25. The blue cluster of languages had two types of inconsistencies, R-H or/and H-R, and their total number was between 1 and 13. 

In the case of single Regulation (EU) No. 2019/1381, languages were divided into two clusters, with only EL included in the second cluster.

Figure A3 in Appendix B shows the division into clusters in terms of the number of inconsistencies identified solely by us using the exact translations of “risk” and “hazard”. The result was similar to the classification in Figure 9. The clustering of the different EU members did not result in a particular group of countries but showed that the same translation problems exist in all countries. However, EL was shown in a separate cluster because only this language had the combination H-SD (35.9%) and H-RF (61.5%), and the proportion did not differ between the new (brown) and the old (blue) regulation (Figure 9). ES was an exception and was positioned in the blue cluster but with a separate branch for the old regulation, which only had a proportion of RA-RD (72.7%) and a small proportion of H-R (27.3%). The same is true for ES in the new regulation (RA-RD 92.9%, H-R 7.1%). The low proportion of H-R is the reason why it was positioned in the blue cluster.

## 4. Discussion

To the best of our knowledge, no study has yet been carried out to investigate the inconsistencies in the translation of “hazard” and “risk” in EU food legislation. With the presented analysis of selected EU regulations dealing with food safety, we focus on the inconsistencies in the translations of “hazard” and “risk”, opening the possibility of harmonising EU regulations even at a higher level. The program HA-RI and the automatic search for inconsistencies, the prepared lists of inconsistencies and their assessment represent efficient tools for the evaluation and harmonisation of the EU legislation.

The number of inconsistencies (the number agreed upon between us and national food experts) decreased significantly when comparing the “old” and “new” regulations, from 336 to 267 (Table 5). However, some problems remain. Some countries have successfully reduced the number of inconsistencies between old and new legislation (e.g., LT, ET), while others remained the same or nearly the same (e.g., EL, PT) or even increased them (e.g., ES). Problems in translation may arise from insufficient knowledge on the part of the translators but also from the lack of appropriate terminology in the national languages. It is expected that these problems will be overcome over time. A good example is the latest Regulation 2019/1381, where relatively few inconsistencies in translations were found, indicating that the new terminology has already taken hold.

Overall, Lithuanian (LT) showed a high number of inconsistencies. Most of them were related to the interchangeability of “hazard” with “risk”. The same interchange was frequently observed in languages that scored high in the total number of inconsistencies, such as in Slovene (SL), Polish (PL), Danish (DA), Romanian (RO) and Portuguese (PT) (Table 5, Figure 4). The same languages, with the exception of Lithuanian and Polish, revealed inconsistencies related to the interchangeability of “risk” with “hazard” Table 5, Figure 5. 

Another language that showed an overall high score was Greek. The issues in the Greek language (EL) were more related to the fact that “hazard” was carried over either as a “source of danger” (πηγή κινδύνου) in Regulation No. 178/2002, which also contains the section defining the terms, and in Regulation No. 852/2004, or as a “risk factor”, which was so defined and used in Regulation No. 625/2017. In Regulation No. 625/2017, the same term “hazard” is defined differently and used in this form throughout the regulation, although it is the same exact term that was already defined in Regulation No. 178/2002. The new term for “hazard” in this newer 2017 regulation is “risk factor” (“παράγοντας κινδύνου”). Besides defining the term differently across regulations, which could lead to confusion and misunderstanding, it is worth mentioning that while “παράγοντας επικινδυνότητας” (where “παράγοντας” means factor and “κινδύνου” means hazard) is meant to be “risk factor”, a more appropriate structure could be “παράγοντας πικινδυνότ” (where “παράγοντας” means factor and “επικινδυνότητας” means risk).

In the case of Spanish, the problem started already with the translation of “risk assessment” in Regulation No. 178/2002. “Risk assessment” is defined as “determinación del riesgo” or “risk determination” instead of the more appropriate expression “evaluacíon de riesgo”. Most food experts involved agreed that the translation does not reflect the intended meaning and is open to interpretation. An accurate translation is available, and therefore there is no reason for inconsistencies. In the languages of the Romance family, “hazard” is translated with a word more equivalent to “danger” (e.g., “danger” in French, “peligro” in Spanish, “perigo” in Portuguese, “pericolo” in Italian, “pericol” in Romanian). The countries speaking these languages joined the EU as follows: 1957—France, Italy, 1986—Spain, Portugal and 2007—Romania. There is a difference of 29 years between the accession of France and Italy and the accession of Spain and Portugal. There is another difference of 21 years between the accession of Spain and Portugal and the accession of Romania. A slight increase in translation inconsistencies was noted, with older Romance-speaking member states having fewer inconsistencies than newer members: France and Italy, compared to Portuguese and Romanian, with Spanish standing out. 

The percentage inconsistency rate showed that, in general, “hazard” interchanges (H-R + H-RF + H-SD) are more frequent in the old and new versions of Regulations EU No. 2002/178, 2004/852, 2004/853 and 2017/625 (maximum values 55.7% and 57.1% in Greek) than R-H interchange rates (maximum values 13.8% in Lithuanian and 6.1% in Polish). The single Regulation (EU) No. 2019/1381 had a significantly lower inconsistency rate. These results suggest that the translation of “hazard” is rather difficult in most languages. Furthermore, as only a proportion of the translations are incorrect, the results suggest inconsistency and/or ignorance on the part of translators rather than a lack of appropriate translations in the national languages.

The results suggest that inconsistencies in the translation of “hazard” and “risk” exist in all EU countries at the level of food regulations. They may therefore be a potential source of confusion for stakeholders, who should be able to distinguish between the two terms in order to be aware of and actively engage in the food safety issue. 

## 5. Conclusions

In summary, food regulations translated from English into other official EU languages constantly interchange “hazard” with “risk” and vice versa. In a few cases, “hazard” was replaced by “source of danger” or “risk factor” and “risk assessment” with “risk determination”. Furthermore, the results have shown that the translation of food regulations and the subsequent corrections depend on the selected group of experts who undertake this task.

It is strongly recommended that risk managers, with the support of risk assessors, adequately train staff responsible for translating food regulations at the national level for each EU Member State. Terminology should be clearly explained and always used as defined. Definitions should be re-evaluated and corrected as necessary. These measures can lead to better harmonisation of food regulations in the EU and, thus, to a better understanding of food safety. It is also advisable to correct all existing food regulations that contain this type of terminology and ensure the correctness of future documents.

## Figures and Tables

**Figure 1 foods-12-02857-f001:**
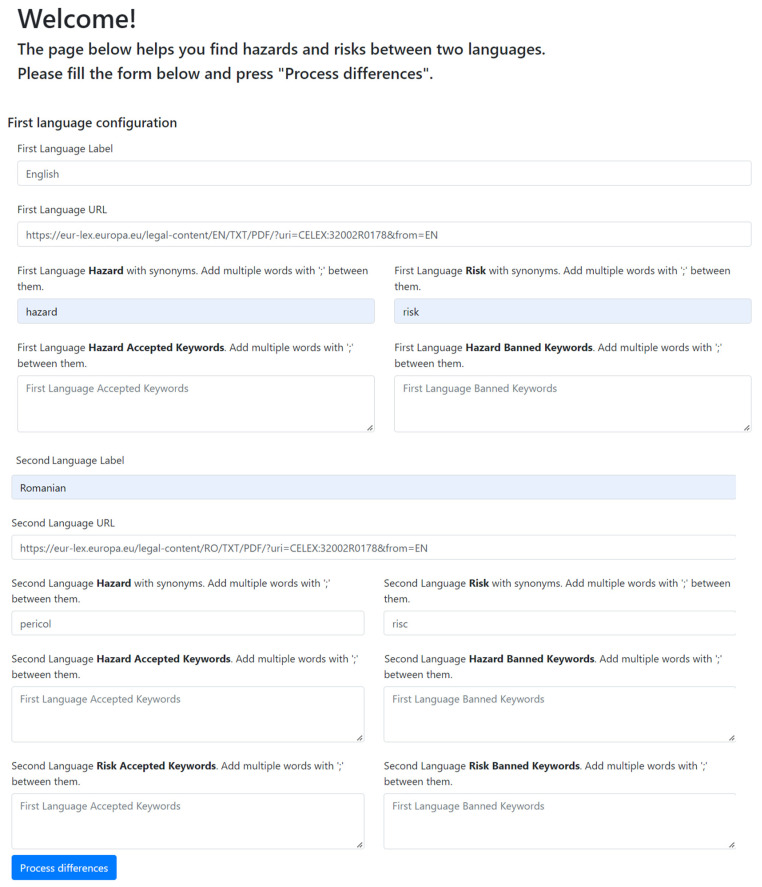
Inputs for the analysis of inconsistencies in Regulation (EU) 2002/178, English vs. Romanian.

**Figure 2 foods-12-02857-f002:**
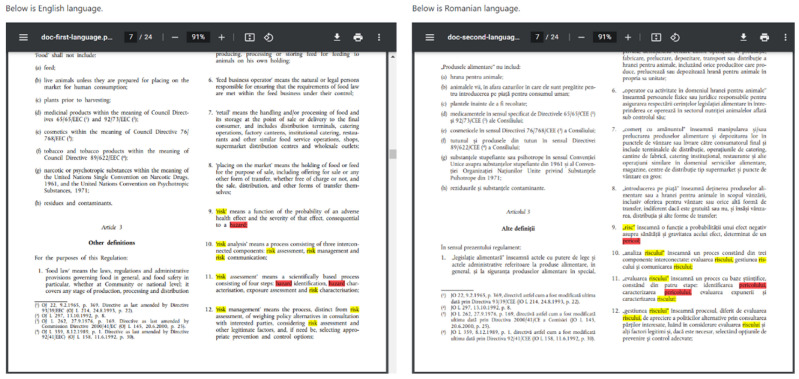
Results after the analysis of inconsistencies in Regulation (EU) 2002/178, English vs. Romanian.

**Figure 3 foods-12-02857-f003:**
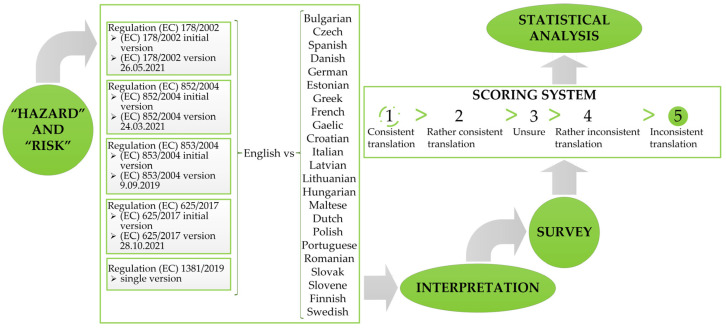
The scheme of the procedure consisted of six steps: (1) selection of terms, in this case, “hazard” and “risk”, (2) selection of regulations, (3) selection of languages and sending invitations to participate to the national food experts, (4) interpretation or search for discrepancies, (5) survey: drawing up lists of discrepancies related to the selected languages and sending the corresponding list to the national expert, and finally performing (6) statistical analysis of the collected fulfilled surveys and conclusions.

**Figure 4 foods-12-02857-f004:**
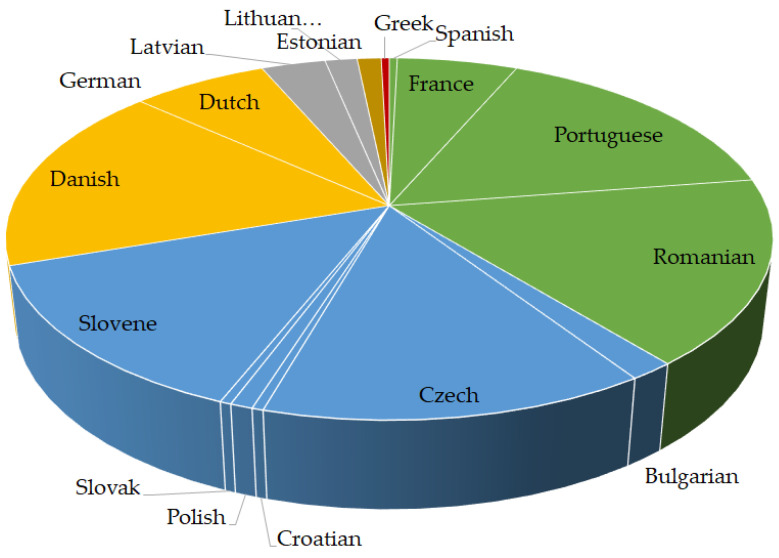
Language-specific proportions of expert-confirmed inconsistencies in Regulations (EU) 2002/178, 2004/852, 2004/853, 2017/625 and 2019/1381 where “hazard” was interchanged with “risk”. Legend: Hellenic (red), Italic (green), Slavic (blue), Germanic (orange), Baltic (grey), Uralic (brown).

**Figure 5 foods-12-02857-f005:**
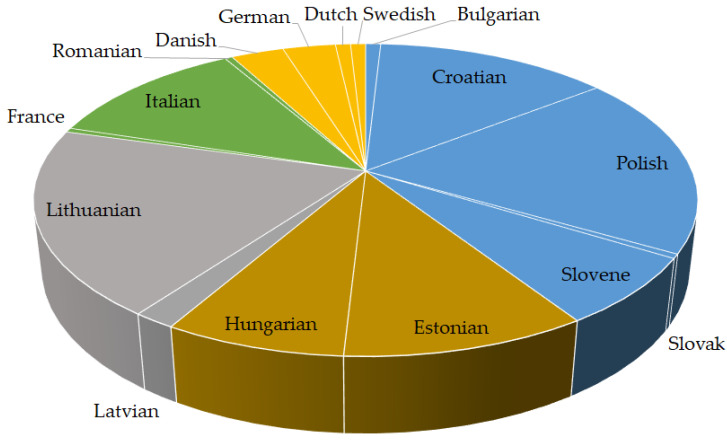
Language-specific proportions of expert-confirmed inconsistencies in Regulations (EU) 2002/178, 2004/852, 2004/853, 2017/625 and 2019/1381 where “risk” was interchanged with “hazard”. Legend: Slavic (blue), Uralic (brown), Baltic (grey), Germanic (orange), Italic (green).

**Figure 6 foods-12-02857-f006:**
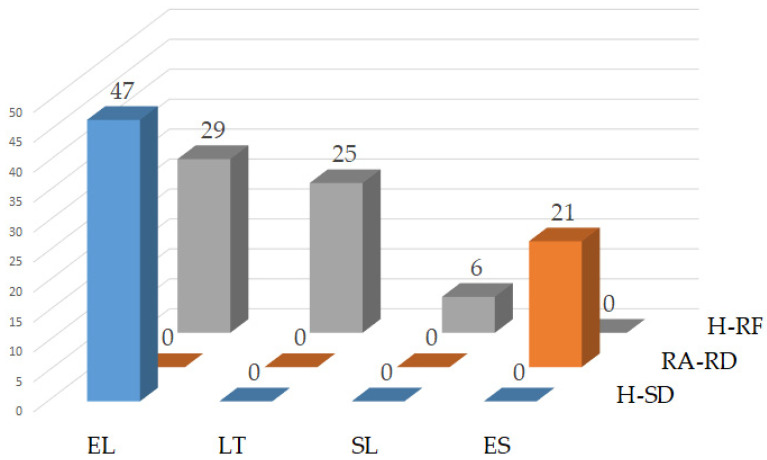
Language-specific proportions of expert-confirmed inconsistencies in Regulations (EU) 2002/178, 2004/852, 2004/853, 2017/625 and 2019/1381 where “hazard” was interchanged with “source of danger” or “risk factor” and “risk assessment” with “risk determination”.

**Figure 7 foods-12-02857-f007:**
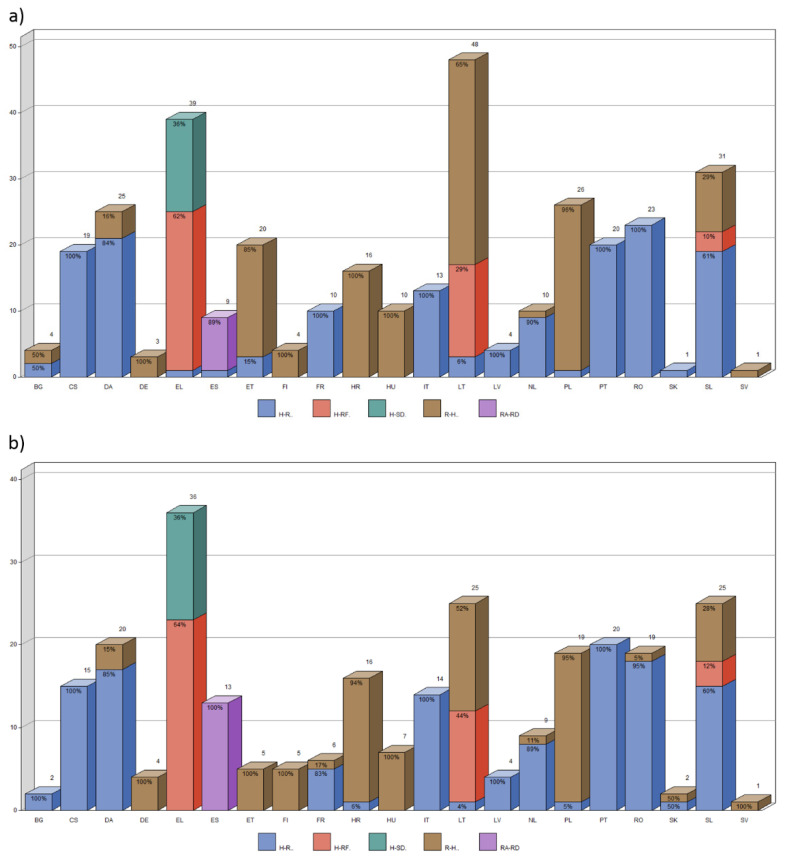
The number of matches between our and the experts’ ratings of inconsistencies for a given language, the proportion of the type of inconsistencies: H-R (blue), H-RF (red), H-SD (green), R-H (brown), and RA-RD (purple) and with respect to the old version (**a**) and new version (**b**) regulations. In the case of multiple experts’ ratings, the average rating of all experts in the respective language group was calculated. The short names of the languages are listed in Table 5.

**Figure 8 foods-12-02857-f008:**
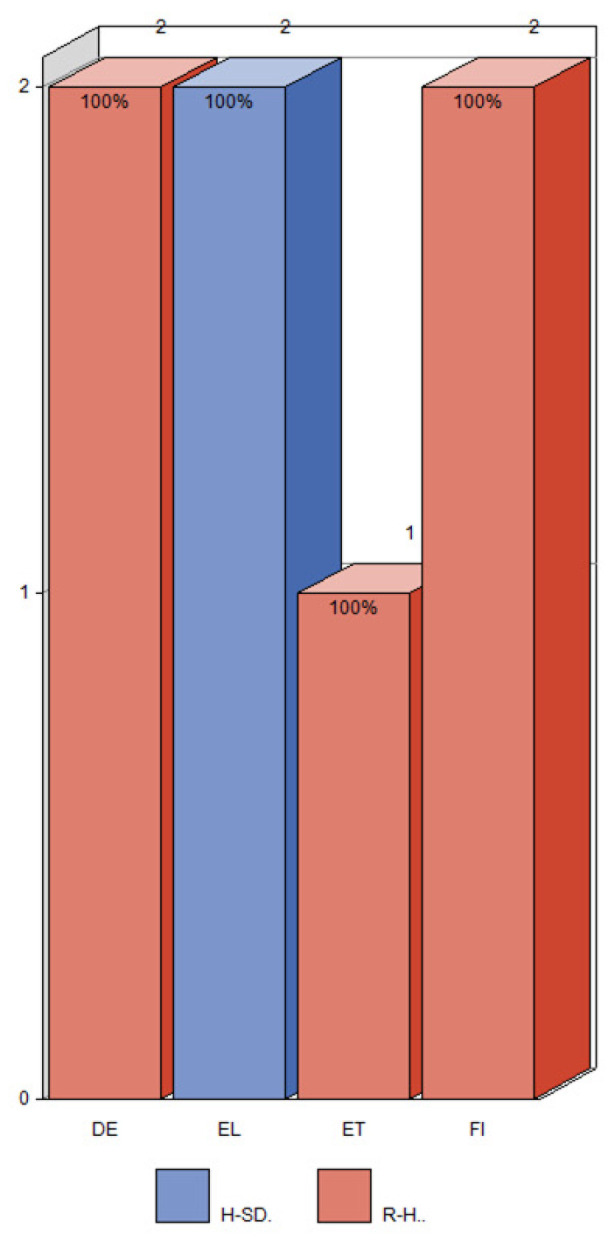
The number of agreements between our and the experts’ ratings of inconsistencies for a given language, the proportion of the type of inconsistencies: H-SD (blue) and R-H (red) and with respect to the single Regulation (EU) No. 2019/1381. In the case of multiple expert ratings, the average rating was calculated for all experts of the respective group. The short names of the languages are listed in Table 5.

**Figure 9 foods-12-02857-f009:**
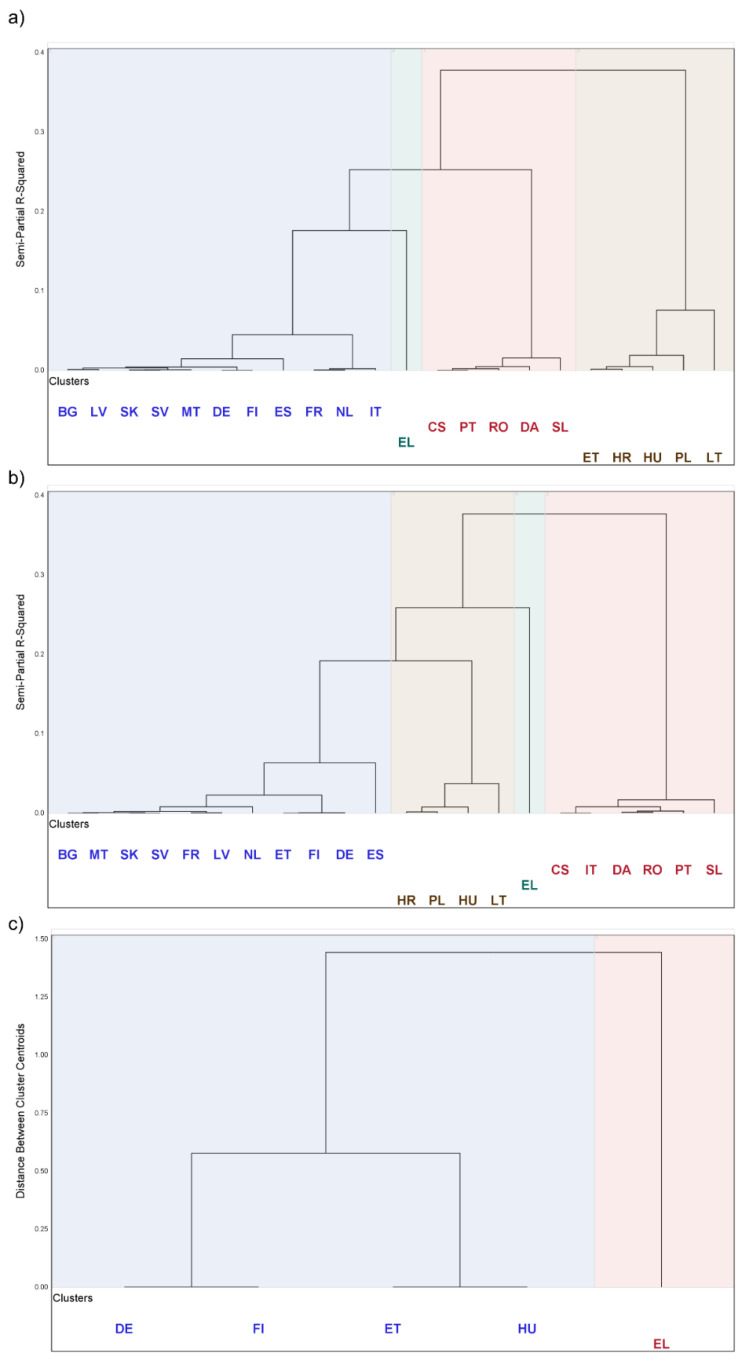
Clustering of languages with respect to the number of R-H, H-RF, RA-RD, H-SD and H-R inconsistencies agreed between us and the experts in (**a**) old Regulations EU No. 2002/178, 2004/852, 2004/853 and 2017/625 (Ward method of clustering), (**b**) new Regulations EU No. 2002/178, 2004/852, 2004/853 and 2017/625 (Ward method of clustering) and (**c**) single Regulation EU No. 1381 (average method of clustering). The short names of the languages are listed in Table 5. Each cluster has its own colour, which is blue, green, red and brown in the case of four clusters and blue and red in the case of only two clusters.

**Table 1 foods-12-02857-t001:** Different combinations of terms for the inconsistencies.

Short Name (Code)	Regulation in English	Translation into National Language
R-H (1 or T1 or T1a)	risk	hazard
H-RF (2 or T2 or T2a)	hazard	risk factor
RA-RD (3 or T3 or T3a)	risk assessment	risk determination
H-SD (4 or T4 or T4a)	hazard	source of danger
H-R (5 or T5 or T5a)	hazard	risk

T1, T2, T3, T4, and T5, were used for the agreed number of inconsistencies between experts and authors, while T1a, T2a, T3a, T4a, and T5a for the number of inconsistencies detected by authors in Appendix A.

**Table 2 foods-12-02857-t002:** Scoring system.

Score	Legend
1	consistent translation
2	rather consistent translation
3	unsure
4	rather inconsistent translation
5	inconsistent translation

**Table 3 foods-12-02857-t003:** Subfamilies/groups of languages of the EU.

Indo-European Family			
Subfamily (Language Group No.)	Division	Subdivisions	Languages
Germanic (Language Group 1)	West Germanic	Anglo-Frisian	English
		Netherlandic-German	German, Dutch
		East-Scandinavian	Swedish, Danish
Italic (Language Group 2)	Romance/Latin	Ibero-Romance	Spanish, Portuguese
		Gallo-Romance	French
		Italo-Dalmatian	Italian
		Eastern Romance	Romanian
Slavic (Language Group 3)	South Slavic	Western	Croatian, Slovene
	West Slavic	Eastern	Bulgarian
		Czech-Slovak	Slovak, Czech
		Lekhitic	Polish
Baltic (Language Group 4)	-	-	Lithuanian, Latvian
Uralic (Group Language 5)	Finno-Ugric	Finnic	Finnish, Estonian
		Ugric	Hungarian
Celtic (Language Group 6)	Insular	Goidelic	Gaelic
Hellenic (Language Group 7)	-	-	Greek
Semitic Family			
Afro-Asiatic/Afrasian/Hamito-Semitic/Semito-Hamitic/Erythraean (Language Group 8)	Semitic	Central-Semitic	Maltese

**Table 4 foods-12-02857-t004:** Accession of member states to the EU.

European Union			
Year of EU Adherence (Accession)	Year Difference	Years of Membership	Country/Countries
1957 (Accession 1)	-	65	France, Italy, Germany, Belgium, Luxembourg, The Netherlands
1973 (Accession 1)	29	49	Denmark, Ireland, United Kingdom (Brexit 2020)
1981 (Accession 2)	21	41	Greece
1986 (Accession 2)	5	36	Spain, Portugal
1995 (Accession 2)	9	27	Austria, Finland, Sweden
2004 (Accession 3)	9	18	Czech Republic, Estonia, Cyprus, Latvia, Lithuania, Hungary, Malta, Poland, Slovakia, Slovenia
2007 (Accession 3)	3	15	Romania, Bulgaria
2013 (Accession 3)	6	9	Croatia

**Table 5 foods-12-02857-t005:** The total number of inconsistencies identified by experts and us.

	Old Version		New Version		Single Version	
Language (Short Name)	Experts (Confirmed Inconsistencies)	Our Determination of Inconsistencies	Experts (Confirmed Inconsistencies)	Our Determination of Inconsistencies	Experts (Confirmed Inconsistencies)	Our Determination of Inconsistencies
Bulgarian (BG)	4	10	2	3		
Czech (CS)	19	19	15	15		
Spanish (ES)	9	11	13	14		
Danish (DA)	25	29	20	20		
German (DE)	3	3	4	4	2	2
Estonian (ET)	20	20	5	5	1	1
Greek (EL)	39	39	36	36	2	2
French (FR)	10	10	6	6		
Croatian (HR)	16	16	16	16		
Italian (IT)	13	13	14	14		
Latvian (LV)	4	6	4	6		
Lithuanian (LT)	48	48	25	25		
Hungarian (HU)	10	14	7	16	0	1
Maltese (MT)	-	5	-	4		
Dutch (NL)	10	10	9	9		
Polish (PL)	26	26	19	19		
Portuguese (PT)	20	20	20	20		
Romanian (RO)	23	23	19	19		
Slovak (SK)	1	2	2	4		
Slovene (SL)	31	31	25	25		
Finnish (FI)	4	4	5	5	2	2
Swedish (SV)	1	2	1	3		
Sum	336	361	267	288	7	8

## Data Availability

The data presented in this study are available on request from the corresponding author.

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
