# Peer review of "Food Risk Analysis: Towards a Better Understanding of “Hazard” and “Risk” in EU Food Legislation"

_foods, 2023, doi:10.3390/foods12152857_

Round 1

Reviewer 1 Report

Attached

Author Response

Dear Reviewer,
Please find my response in the attached WORD file.
Best regards, Tomaz

Reviewer 2 Report

The Authors presenting the study on Food Risk Analysis: Towards a Better Understanding of „Hazard” and „Risk” in EU food legislation", applied a novel approach to describe a relevant aspect in the field of communication and in particular to point the need for further improvement in this respect. 

INTRODUCTION section: I would advice to include briefly  that the terms Hazards and Risk are used also in other domains and legislation (i.e. REGULATION (EU) 2016/429)  and this methodology could be applied also to them. 

M&M section: please provide a more detailed explanation on the expert involvelment (i.e. total number of expert/by country/per language involved in the process) and on the way harmonization in the screening, applied to all the countries, was ensured.

DISCUSSION section: is it possible to estimate the impact of these inconsistencies and the main target affected (i.e policy-makers?), when considering the activities performed at Member state level? This is a curiosity from my side but might be interesting to include, if there is any evidence.

In general, the Authors described with a novel approach an important issue known by the Risk assessment community but this research provide evidence on the magnitude of the issue itself for action to be taken to address it.

Author Response

(The authors gave the same response as above.)

Reviewer 3 Report

The article confirms that differences in the translation of risk-related terms in regulations written in English exist when they are translated into the languages of the member countries.

I fully agree with the authors that this fact can create confusion. However, I think it is essential to differentiate between inconsistencies that are errors, such as confusing "hazard" with "risk", and inconsistencies that are equivalent or synonymous terms, which, although they could lead to confusion, are not semantically incorrect, such as speaking of "risk determination", when what is to be done is to determine or obtain the exact value of the risk.

In this sense, as a reader of this article, It would be interesting to introduce a table with the most appropriate translation of each term in the different languages (with the percentage in which it has been used) and what other terms or inconsistencies have been used (with the percentage of each of them).

In addition, the conclusions only refer to the terms Hazard and Risk and should be concluded for all the terms studied.

Author Response

(The authors gave the same response as above.)

Round 2

Reviewer 3 Report

None